# TD Learning with Neural Networks - Study of the Leakage Propagation Problem

**Hugo Penedones, Damien Vincent, Timothy Mann, Sylvain Gelly**
DeepMind & Google Brain
London, UK & Zurich, Switzerland
{hugopen, damienv, timothymann, sylvaingelly}@google.com

## Abstract

In On-Policy Evaluation, one estimates the value function of the data-generating policy with algorithms like Monte-Carlo regression (MC) or Temporal-Difference Learning (TD). We investigate the issue of poor estimation when using a function approximator like a neural network, due to limited data, limited capacity or training process, and how approximation errors can be further propagated by TD bootstrap updates. We suggest that this problem may be mitigated by first learning (unsupervisedly) a representation that separates states that look similar, but are actually quite distant when one looks at the trajectories followed by the policy.

## 1 Introduction

Reinforcement Learning (Sutton & Barto, 1998) deals with the setup where an agent is interacting with an environment: at each time step it sees the current *state* (or just an *observation*) and takes one *action*, which will determine the immediate *reward* $R_t$ it will get and the next state.

Typically, the agent is interested in maximizing the *expected return*, where the return $G_t$ is often a discounted sum of all rewards in the trajectory. The *state-value function*, $v_\pi$, is defined as the expected return, when the agent starts in a given state $s$ and then behaves according to the policy $\pi$.

$$v_\pi(s) \doteq \mathbb{E}_\pi[G_t | S_t = s]$$

Estimating the state-value function for the policy that collected the data is the problem known as *On-Policy Evaluation*, in contrast with the (harder) problem of *Off-Policy Evaluation* in which one tries to estimate the value function of a different policy (Precup et al., 2001; Dann et al., 2014).

If our value function approximation $\hat{v}_\pi$ is parameterised by $\mathbf{w}$ (e.g. neural network weights), and $v_\pi$ is the ground truth value function, the goal is to minimize the Mean-Squared Value Error (MSVE):

$$MSVE(\mathbf{w}) \doteq \mathbb{E}\big[(v_\pi(s) - \hat{v}_\pi(s, \mathbf{w}))^2\big] \ . \tag{1}$$

### 1.1 Environments

We run our experiments in a toy environment in which an agent navigates a two-dimensional space taking fixed-length steps. The state is represented by the real-valued $(x, y)$ coordinates. There might be walls, which the agent can not cross, and one or more circle-shaped areas that give a +30 reward (Detailed setup given in Appendix A). We collect data using a random policy initialized at random starting locations and we set a fixed rollout length limit. Precise estimates of the ground truth value function are shown in Figure 1. We do this by simulating in each point of an uniform grid, 1000 rollouts and averaging their Monte-Carlo returns.

### 1.2 The *leakage propagation* problem

Neural networks, being parametric models with limited capacity, are forced to make approximation trade-offs. In some areas of the state space, we might have sharp discontinuities in the (true) value

function, but our neural network function approximator might be a bad estimator in those areas, due to limited amount of data, limited capacity, or the training process.

Such limitations can degrade the accuracy of our value function estimation, in both Monte-Carlo Regression and Temporal-Difference Learning. However note that in TD-Learning the problem can be aggravated by an effect of *leakage propagation*. This can happen because TD bootstraps its estimates from other estimates, and the error is iteratively propagated to surrounding states. See Figure 1 for an illustration of this effect in TD Learning.

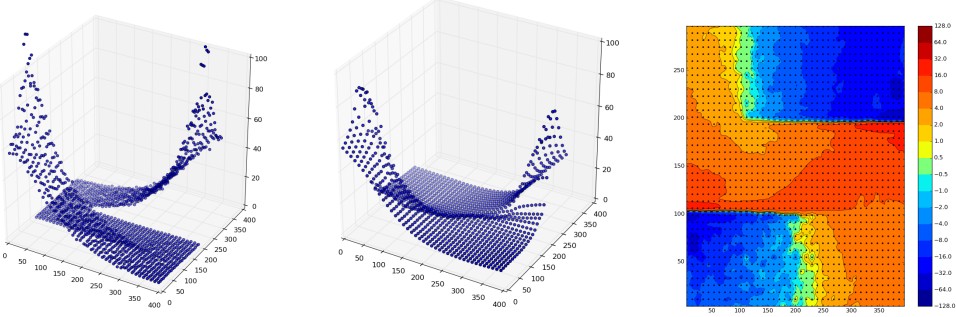

Figure 1: Ground truth value function (left). Predictions of TD learned value function (middle) and its error heatmap compared to ground truth (right). Notice the leakage propagation from across the walls in the middle section, which is being overestimated (red colors).

## 2 REPRESENTATIONS FOR ON-POLICY EVALUATION

We study the setup where we first find an intermediate state representation, and then learn the value function on top of it. This intermediate embedding is used to learn the topology of the trajectories. This does not require any reward information and can be used to bias the value function approximator towards solutions that extrapolate and generalize well. In particular, if two states have very similar input features but are not connected (or took a large number of steps to connect), then the intermediate embedding should push those two states apart so the value network does not extrapolate from one to the other, by continuity or smoothness.

We consider three different embeddings. The first one is a hand-crafted transformation (also referred to as "oracle embedding") that separates nearby points across the walls perfectly, assuming privileged knowledge about the map layout. The second embedding is given by:

$$\Psi_\pi(s) = \mathbb{E}_\pi \left[ \sum_t \gamma^t \phi(S_t) | S_0 = s \right]$$

also known as successor features (SF), a concept introduced in Barreto et al. (2016) and Kulkarni et al. (2016) as an extension of the successor representation (Dayan, 1993) for a continuous state space. In the toy problem, $\phi(s) = s$, $\Psi$ then corresponds to the average discounted position of the agent in the labyrinth. Finally, the last embedding is derived by optimizing what we call the "rubberband loss". This loss includes a term which tries to keep states within the same trajectory at a distance roughly proportional to the number of steps in between them (cohesion) and an expansion term which pushes away any pair of states. An example of those 3 embeddings is given by Figure 2.

## 3 EXPERIMENTS

In all our experiments we use a Multi-Layer Perceptron (MLP) decomposed into two sets of layers: 1) the embedding network with layers of sizes: [20, 20, 20, 2] and the value network with sizes [30, 30, 1]. We fixed the output of the embedding network to always be 2-dimensional, so that we could easily visualize the learned embedding and compare it to our hand-designed embeddings.

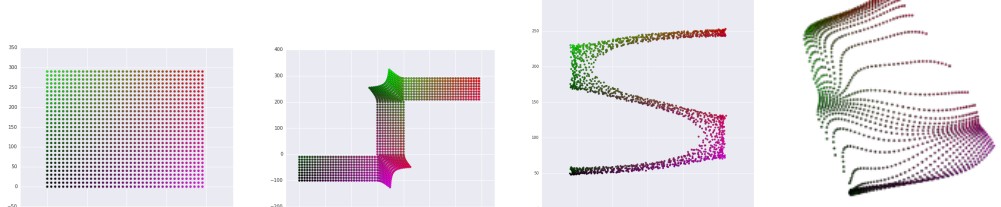

Figure 2: Embeddings used in the toy problem. From left to right: (a) Original state space, (b) Oracle embedding, (c) Ground truth SF embedding, (d) Estimated rubberband embedding.

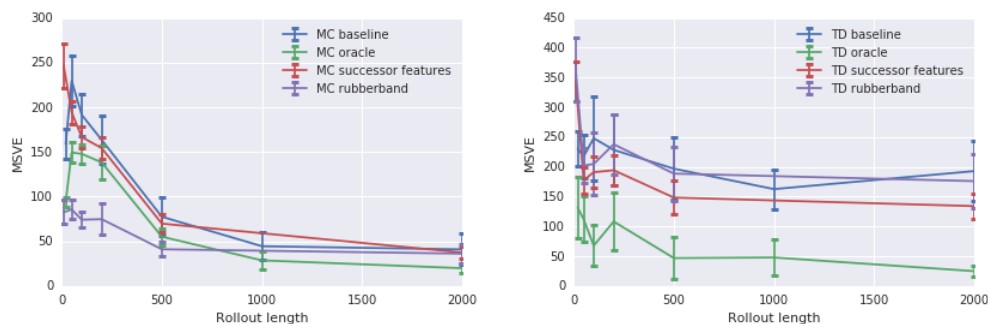

Figure 3: MSVE comparison of various embeddings with different amounts of training data, for MC (left) and TD (right). Baseline refers to the optimization without constraint on the embedding.

The experiments in Figure 3 show that the hand-designed intermediate representation leads to a significantly better value-function, with reduced MSVE. Furthermore, we show that the unsupervised learning strategies can also get us partway there, which opens the door for further investigations.

## 3.1 RELATED WORK

Temporal-Difference Learning with Function Approximation has been analysed by Tsitsiklis & Van Roy (1997), with convergence results for the Linear case, and showcasing a divergence example for a nonlinear approximator. Some convergence results for the non-linear case were later achieved by (Bhatnagar et al., 2009), but the solution might have poor MSVE. Proto-value functions (Mahadevan, 2005) also try to tackle the problem of having a better representation for learning value functions, but do not use neural embeddings to achieve so. Work on Successor Representations and Features is covered by (Barreto et al., 2016; Kulkarni et al., 2016; Dayan, 1993). Great progress has also been made at improving the stability of Deep Reinforcement Learning with techniques like (Prioritized) Experience Replay (Schaul et al., 2015), Target Networks (Mnih et al., 2015), but these focus more on the setup of *online* and *off-policy* learning for control tasks.

## 4 DISCUSSION AND CONCLUSION

We described the *leakage propagation* problem that can happen in On-Policy Evaluation with Temporal-Difference Learning and neural networks, and tried to quantitatively and visually assess it. We showed that privileged-knowledge representations can mitigate the problem and bring significant estimation accuracy gains; we then suggested ways of using Unsupervised Learning to get partway there. The additional unsupervised losses are simple to implement, and make use of topological information about the trajectories, that neither Monte-Carlo or Temporal-Difference Learning explicitly exploit.

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

# APPENDIX A    TOY PROBLEM SETUP

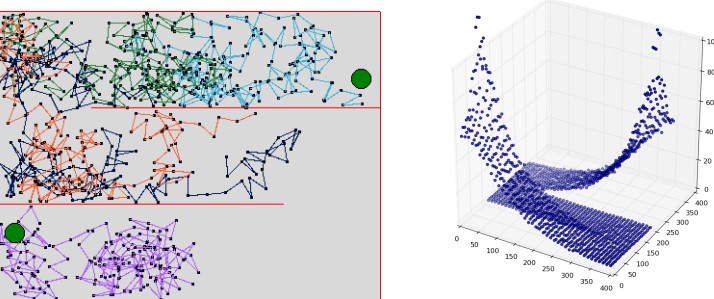

Figure 4: Left: random policy trajectories in a S-shaped layout with two reward zones (in green). Right: ground truth value function.

