# OpenReview forum: "TD Learning with Neural Networks - Study of the Leakage Propagation Problem"
_ICLR.cc/2018/Workshop — Reject_

### Official Review · AnonReviewer2 · 2018-03-06
**Interesting work, but lacks formal/theoretical analysis and connection to a broader community**

**Rating:** 6
**Confidence:** 4

**Review:**

Pros: 1. Investigated an important technical problem in RL
          2. empirical study is limited, but convincing
Cons: 1. Lacking formal definition and analysis of leakage propagation. Under what the condition can leakage propagation happen? This seems to be a general problem for TD-learning in state space with singularities.
         2. lacking connection to a broader community. Although solving different problems, people have considered similar ideas of first perform unsupervised learning (dimension reduction/representation), and then perform planning/reinforce learning on embedding spaces, e.g.,
Chen et al. Motion Planning with Diffusion Maps (IROS16)
Marlos et al. A Laplacian Framework for Option Discovery in Reinforcement Learning (ICML17)

---

> ### Author Response · Authors · 2018-03-23
> **Thank you. All fair points and good suggestions!**
>
> Thank you very much for your fair assessment of the strengths and limitations of our work. In an extended version of this paper we will expand on the connections to the related literature, including the ones you mentioned.
>
> We agree that it would be great to have a more formal definition of the leakage propagation. At the moment we can quantitatively measure the MSVE of the value function estimation, but we can not exactly pinpoint what fraction of that error comes from the leakage propagation, and what fraction comes from other factors (limited capacity of network, issues in the optimization process, etc.). The closest we got to that was to first train with MC and then continue training with TD.
> It would also be great to understand under what exact conditions this leakage propagation happens, indeed. We were able to design environments where it is more likely to happen (continuous state spaces having sharp discontinuities in the value function), but it's not yet a very precise characterization.
> Perhaps we can make progress in those directions with future work. Thanks for your contribution.

---

### Official Review · AnonReviewer3 · 2018-03-08
**nothing new presented**

**Rating:** 2
**Confidence:** 5

**Review:**

The paper considers the idea of projecting samples from an RL task into an intermediate representation before performing TD learning with a neural network.  It is demonstrated that the quality of learned value functions depends in part upon this representation, as error propagates along states that are nearby in the chosen representation.

It is well-known and -researched that representation and feature selection is an important step in RL.  A great deal of prior work is ignored, to the extent that a list would be unwieldy.  The paper provides some useful visuals for an introductory discussion on feature selection in RL, but otherwise no real insights or surprises.

---

> ### Author Response · Authors · 2018-03-23
> **Thank you, more detailed feedback welcome**
>
> Thanks for your review, despite the rejection. Do you mind sharing just the 3 most relevant related works, in your opinion, that we failed to acknowledge? We see this as a great opportunity to benefit from your expertise and constructive feedback, which is arguably the best possible outcome of the peer review system. Greatly appreciated.

---

### Official Review · AnonReviewer1 · 2018-03-10
**Addressing an important problem, but the content is not clear and precise enough**

**Rating:** 5
**Confidence:** 3

**Review:**

The papers empirically studies the leakage propagation in TD learning, which is the problem of propagation of error through the state space caused by the bootstrapping of TD.
It suggests that the use of proper state space embedding might alleviate this problem. If two states are nearby in the input representation, but are far according to the dynamics of the MDP (that is, it takes many steps to reach from one state to another), the paper then suggests that the embedding should map them to two afar points.
The paper considers three embeddings, one of them manually designed and two others learned by an unsupervised signal, and shows that they reduce the leakage propagation problem.

The paper addresses an important problem, but I believe this work requires more study and deliberation (understanding that this is a workshop paper). Currently I cannot be very enthusiastic about this paper. For example, some of the shortcomings of this work are:

- The paper is not clear about the choices of embedding. What is the hand-crafted transformation, and what exactly is the rubber band loss? It is desirable to be more formal here.

- Much of the intuition of this work is related to the porto-value functions and the eigenfunctions of Laplace-Beltrami operator, but the paper is only cursorily mention that (Mahadevan, 2005) did not use neural embedding to achieve so. Even so that is the case, the relation deserves much better comparison and acknowledgement.

- It is stated in Section 1.1 that the detailed setup is given in Appendix A, but that appendix only has a figure. What is the dynamics of the movement? Is it stochastic or deterministic?

---

> ### Author Response · Authors · 2018-03-23
> **Thank you, we agree with all the points.**
>
> Thank you very much for this accurate description of the strengths and the shortcomings of this short paper.
> We acknowledge that, unfortunately, in the process of reducing our draft to fit the 3 pages limit for the workshop, we ended up removing too much detail about the different embeddings (notably the rubber band loss and the hand-crafted transformation), as well as the detailed descriptions of the environments. We will naturally address those with an extended version of the paper, which can then also contain deeper comparisons with related work (notably proto-value functions, which indeed was only superficially mentioned).
>
> Note: the environment is deterministic. It accepts every step from the agent, as long as it does not collide against a wall. The policy we used is stochastic, though (it picks a random direction to move towards, at each point in time).

---

> > ### Comment · AnonReviewer1 · 2018-03-27
> > **Thank you!**
> >
> > Thank you for your reply. I hope to see the improved paper published in the near future.

---

### Decision · Program_Chairs · 2018-03-20
**ICLR 2018 Workshop Acceptance Decision**

**Decision:**

Reject

**Comment:**

Based on the reviews, this paper has not been accepted for presentation at the ICLR workshop. However, the conversation and updates can continue to appear here on OpenReview.